# Efficient Smooth Non-Convex Stochastic Compositional Optimization via Stochastic Recursive Gradient Descent

**Wenqing Hu**
Missouri University of Science and Techology
huwen@mst.edu

**Chris Junchi Li**✉
Tencent AI Lab
junchi.li.duke@gmail.com

**Xiangru Lian**
University of Rochester
admin@mail.xrlian.com

**Ji Liu**
University of Rochester & Kwai Inc.
ji.liu.uwisc@gmail.com

**Huizhuo Yuan**∗
Peking University
hzyuan@pku.edu.cn

## Abstract

Stochastic compositional optimization arises in many important machine learning applications. The objective function is the composition of two expectations of stochastic functions, and is more challenging to optimize than vanilla stochastic optimization problems. In this paper, we investigate the stochastic compositional optimization in the general smooth non-convex setting. We employ a recently developed idea of *Stochastic Recursive Gradient Descent* to design a novel algorithm named SARAH-Compositional, and prove a sharp Incremental First-order Oracle (IFO) complexity upper bound for stochastic compositional optimization: $\mathcal{O}((n+m)^{1/2}\varepsilon^{-2})$ in the finite-sum case and $\mathcal{O}(\varepsilon^{-3})$ in the online case. Such a complexity is known to be the best one among IFO complexity results for non-convex stochastic compositional optimization. Numerical experiments on risk-adverse portfolio management validate the superiority of SARAH-Compositional over a few rival algorithms.

## 1 Introduction

We consider the general smooth, non-convex compositional optimization problem of minimizing the composition of two expectations of stochastic functions:

$$\min_{x \in \mathbb{R}^d} \quad \{\Phi(x) \equiv (f \circ g)(x)\}, \tag{1}$$

where the outer and inner functions $f : \mathbb{R}^l \to \mathbb{R}, g : \mathbb{R}^d \to \mathbb{R}^l$ are defined as $f(y) := \mathbb{E}_v[f_v(y)]$, $g(x) := \mathbb{E}_w[g_w(y)]$, $v$ and $w$ are index random variables, and each component $f_v$, $g_w$ are smooth but *not* necessarily convex. Compositional optimization can be used to formulate many important machine learning problems, e.g. reinforcement learning (Sutton and Barto, 1998), risk management (Dentcheva et al., 2017), multi-stage stochastic programming (Shapiro et al., 2009), deep neural net (Yang et al., 2019), etc. We list a specific application instance that can be written in the stochastic compositional form of (1), namely the *risk-adverse portfolio management problem*, formulated as

$$\min_{x \in \mathbb{R}^N} \quad -\frac{1}{T}\sum_{t=1}^{T}\langle r_t, x\rangle + \frac{1}{T}\sum_{t=1}^{T}\left(\langle r_t, x\rangle - \frac{1}{T}\sum_{s=1}^{T}\langle r_s, x\rangle\right)^2, \tag{2}$$

---

∗Partial work was performed when the author was an intern at Tencent AI Lab. Full version of this paper is available at: http://arxiv.org/abs/1912.13515

| Algorithm | Finite-sum | Online |
|---|---|---|
| SCGD (Wang et al., 2017a) | unknown | $\varepsilon^{-8}$ |
| Acc-SCGD (Wang et al., 2017a) | unknown | $\varepsilon^{-7}$ |
| SCGD (Wang et al., 2017b) | unknown | $\varepsilon^{-4.5}$ |
| SCVR / SC-SCSG (Liu et al., 2017) | $(n+m)^{4/5}\varepsilon^{-2}$ | $\varepsilon^{-3.6}$ |
| VRSC-PG (Huo et al., 2018) | $(n+m)^{2/3}\varepsilon^{-2}$ | unknown |
| **SARAH-Compositional (this work)** | $(n+m)^{1/2}\varepsilon^{-2}$ | $\varepsilon^{-3}$ |

Table 1: Comparison of IFO complexities with different algorithms for general non-convex problem.

where $r_t \in \mathbb{R}^N$ denotes the returns of $N$ assets at time $t$, and $x \in \mathbb{R}^N$ denotes the investment quantity corresponding to $N$ assets. The goal is to maximize the return while controlling the variance of the portfolio. (2) can be written as a compositional optimization problem with two functions

$$g(x) = \left[x_1, x_2, \ldots, x_N, \frac{1}{T}\sum_{s=1}^{T}\langle r_s, x\rangle\right]^{\top}, \tag{3}$$

$$f(w) = -\frac{1}{T}\sum_{t=1}^{T}\langle r_t, w_{\backslash(N+1)}\rangle + \frac{1}{T}\sum_{t=1}^{T}\left(\langle r_t, w_{\backslash(N+1)}\rangle - w_{N+1}\right)^2, \tag{4}$$

where $w_{\backslash(N+1)}$ denotes the (column) subvector consisting of the first $N$ coordinates of $w$, and $w_{N+1}$ denotes the $(N+1)$-th coordinate of $w$.

Compared with vanilla stochastic optimization problem where the optimizer is allowed to access the stochastic gradients, stochastic compositional problem (1) is more difficult to solve. Classical algorithms for solving (1) are often more computationally challenging. This is mainly due to the nonlinear structure of the composition function with respect to the random index pair $(v, w)$. Treating the objective function as an expectation $\mathbb{E}_v f_v(g(x))$, computing each iterate of the gradient estimation involves recalculating $g(x) = \mathbb{E}_w g_w(x)$, which is either time-consuming or impractical. To tackle such weakness in practice, Wang et al. (2017a) firstly introduce a two-time-scale algorithm called Stochastic Compositional Gradient Descent (SCGD) along with its accelerated (in Nesterov's sense) variant Acc-SCGD, and provide a first convergence rate analysis to that problem. Subsequently, Wang et al. (2017b) proposed accelerated stochastic compositional proximal gradient algorithm (ASC-PG) which improves over the upper bound complexities in Wang et al. (2017a). Furthermore, variance reduced gradient methods designed specifically for compositional optimization on non-convex settings arises from Liu et al. (2017) and later generalized to the nonsmooth setting (Huo et al., 2018). These approaches aim at getting variance reduced estimators of $g$, $\partial g$ and $\partial g(x)\nabla f(g(x))$, respectively. Such success signals the necessity and possibility of designing a special algorithm for non-convex objectives with better convergence rates.

In this paper, we propose an efficient algorithm called SARAH-Compositional for the stochastic compositional optimization problem (1). For notational simplicity, we let $n, m \geq 1$ and the index pair $(v, w)$ be uniformly distributed over the product set $[1, n] \times [1, m]$, i.e.

$$\Phi(x) = \frac{1}{n}\sum_{i=1}^{n} f_i\left(\frac{1}{m}\sum_{j=1}^{m} g_j(x)\right). \tag{5}$$

We use the same notation for the online case, in which case either $n$ or $m$ can be infinite.

A fundamental theoretical question for stochastic compositional optimization is the Incremental First-order Oracle (IFO) (the number of individual gradient and function evaluations; see Definition 1 in §2 for a precise definition) complexity bounds for stochastic compositional optimization. Our new SARAH-Compositional algorithm is developed by integrating the iteration of *Stochastic Recursive Gradient Descent* (Nguyen et al., 2017), shortened as SARAH,[2] with the stochastic compositional optimization formulation (Wang et al., 2017a). The motivation of this approach is that SARAH

with specific choice of stepsizes is known to be *optimal* in stochastic optimization and regarded as a cutting-edge variance reduction technique, with significantly reduced oracle access complexities than earlier variance reduction method (Fang et al., 2018). We prove that SARAH-Compositional can reach an IFO computational complexity of $\mathcal{O}(\min\left((n+m)^{1/2}\varepsilon^{-2}, \varepsilon^{-3}\right))$, improving the best known result of $\mathcal{O}(\min\left((n+m)^{2/3}\varepsilon^{-2}, \varepsilon^{-3.6}\right))$ in non-convex compositional optimization. See Table 1 for detailed comparison.

**Related Works** Classical first-order methods such as gradient descent (GD), accelerated gradient descent (AGD) and stochastic gradient descent (SGD) have received intensive attetions in both convex and non-convex optimization (Nesterov, 2004; Ghadimi and Lan, 2016; Li and Lin, 2015). When the objective can be written in a finite-sum or online/expectation structure, variance-reduced gradient (a.k.a. variance reduction) techniques including SAG (Schmidt et al., 2017), SVRG (Xiao and Zhang, 2014; Allen-Zhu and Hazan, 2016; Reddi et al., 2016), SDCA (Shalev-Shwartz and Zhang, 2013, 2014), SAGA (Defazio et al., 2014), SCSG (Lei et al., 2017), SNVRG (Zhou et al., 2018), SARAH/SPIDER (Nguyen et al., 2017; Fang et al., 2018; Wang et al., 2019; Nguyen et al., 2019), etc., can be employed to improve the theoretical convergence properties of classical first-order algorithms. Notably in the smooth nonconvex setting, Fang et al. (2018) recently proposed the SPIDER-SFO algorithm which non-trivially hybrids the iteration of stochastic recursive gradient descent (SARAH) (Nguyen et al., 2017) with the normalized gradient descent. In the representative case of batch-size 1, SPIDER-SFO adopts a small step-length that is proportional to $\varepsilon^2 \wedge \varepsilon n^{-1/2}$ where $\varepsilon$ is the squared targeted accuracy, and (by rebooting the SPIDER tracking iteration once every $n \wedge \mathcal{O}(\varepsilon^{-2})$ iterates) the variance of the stochastic estimator can be constantly controlled by $\mathcal{O}(\varepsilon^2)$. For finding $\varepsilon$-accurate solution purposes, recent works Wang et al. (2019); Nguyen et al. (2019) discovered two variants of the SARAH algorithm that achieve the same complexity as SPIDER-SFO (Fang et al., 2018) and SNVRG (Zhou et al., 2018).[3] The theoretical convergence property of SARAH/SPIDER methods in the smooth non-convex case outperforms that of SVRG, and is provably optimal under a set of mild assumptions (Arjevani et al., 2019; Fang et al., 2018; Nguyen et al., 2019; Wang et al., 2019).

It turns out that when solving compositional optimization problem (1), classical first-order methods for optimizing a single objective function can either be non-applicable or it brings at least $\mathcal{O}(m)$ queries to calculate the inner function $g$. To remedy this issue, Wang et al. (2017a,b) considered the stochastic setting and proposed the SCGD algorithm to calculate or estimate the inner finite-sum more efficiently, achieving a polynomial rate that is independent of $m$. Later on, Lian et al. (2017); Liu et al. (2017); Huo et al. (2018) and Lin et al. (2018) merged SVRG method into the compositional optimization framework to do variance reduction on all three steps of the estimation. In stark contrast, our work adopts the SARAH/SPIDER method which is theoretically more efficient than the SVRG method in the non-convex compositional optimization setting.

**Contributions** This work makes two contributions as follows. First, we propose a new algorithm for stochastic compositional optimization called SARAH-Compositional, which operates SARAH/SPIDER-type recursive variance reduction to estimate relevant quantities. Second, we conduct theoretical analysis for both online and finite-sum cases, which verifies the superiority of SARAH-Compositional over the best known previous results. In the finite-sum case, we obtain a complexity of $(n+m)^{1/2}\varepsilon^{-2}$ which improves over the best known complexity $(n+m)^{2/3}\varepsilon^{-2}$ achieved by Huo et al. (2018). In the online case we obtain a complexity of $\varepsilon^{-3}$ which improves the best known complexity $\varepsilon^{-3.6}$ obtained in Liu et al. (2017).

**Notational Conventions** Throughout the paper, we treat the parameters $L_g, L_f, L_\Phi, M_g, M_f, \Delta$ and $\sigma$ as global constants. Let $\|\bullet\|$ denote the Euclidean norm of a vector or the operator norm of a matrix induced by Euclidean norm, and let $\|\bullet\|_F$ denotes the Frobenius norm. For fixed $T \geq t \geq 0$ let $x_{t:T}$ denote the sequence $\{x_t, ..., x_T\}$. Let $\mathbf{E}_t[\bullet]$ denote the conditional expectation $\mathbf{E}[\bullet|x_0, x_1, ..., x_t]$. Let $[1, n] = \{1, ..., n\}$ and $S$ denote the cardinality of a multi-set $\mathcal{S} \subseteq [1, n]$ of samples (a generic set that permits repeated instances). The averaged sub-sampled stochastic estimator is denoted as $\mathcal{A}_\mathcal{S} = (1/S) \sum_{i \in \mathcal{S}} \mathcal{A}_i$ where the summation counts repeated instances. We denote $p_n = \mathcal{O}(q_n)$ if there exist some constants $0 < c < C < \infty$ such that $cq_n \leq p_n \leq Cq_n$ as $n$ becomes large. Other notations are explained at their first appearances.

**Organization** The rest of our paper is organized as follows. §2 formally poses our algorithm and assumptions. §3 presents the convergence rate theorem and §4 presents numerical experiments that apply our algorithm to the task of portfolio management. We conclude our paper in §5. Proofs of convergence results for finite-sum and online cases and auxiliary lemmas are deferred to §A and §B in the supplementary material.

## 2 SARAH for Stochastic Compositional Optimization

Recall our goal is to solve the compositional optimization problem (1), i.e. to minimize $\Phi(x) = f(g(x))$ where

$$f(y) := \frac{1}{n}\sum_{i=1}^{n} f_i(y), \qquad g(x) := \frac{1}{m}\sum_{j=1}^{m} g_j(x).$$

Here for each $j \in [1,m]$ and $i \in [1,n]$ the functions $g_j : \mathbb{R}^d \to \mathbb{R}^l$ and $f_i : \mathbb{R}^l \to \mathbb{R}$. We can formally take the derivative to the function $\Phi(x)$ and obtain (via the chain rule) the gradient descent iteration

$$x_{t+1} = x_t - \eta[\partial g(x_t)]^\top \nabla f(g(x_t)),\tag{6}$$

where the $\partial$ operator computes the Jacobian matrix of the smooth mapping, and the gradient operator $\nabla$ is only taken with respect to the first-level variable. As discussed in §1, it can be either impossible (online case) or time-consuming (finite-sum case) to estimate the terms $\partial g(x_t) = \frac{1}{m}\sum_{j=1}^{m}\partial g_j(x_t)$ and $g(x_t) = \frac{1}{m}\sum_{j=1}^{m} g_j(x_t)$ in the iteration scheme (6). In this paper, we design a novel algorithm (SARAH-Compositional) based on Stochastic Compositional Variance Reduced Gradient method (see Lin et al. (2018)) yet hybriding with the stochastic recursive gradient method Nguyen et al. (2017). As the readers see later, our SARAH-Compositional is more efficient than all existing algorithms for non-convex compositional optimization.

We introduce some definitions and assumptions. First, we assume the algorithm has accesses to an Incremental First-order Oracle (IFO) in our black-box environment (Lin et al., 2018); also see (Agarwal and Bottou, 2015; Woodworth and Srebro, 2016) for vanilla optimization case:

**Definition 1** (IFO)**.** *(Lin et al., 2018) The Incremental First-order Oracle (IFO) returns, when some $x \in \mathbb{R}^d$ and $j \in [1,m]$ are inputted, the vector-matrix pair $[g_j(x), \partial g_j(x)]$ or when some $y \in \mathbb{R}^l$ and $i \in [1,n]$ are inputted, the scalar-vector pair $[f_i(y), \nabla f_i(y)]$.*

Second, our goal in this work is to find an $\varepsilon$-accurate solution, defined as

**Definition 2** ($\varepsilon$-accurate solution)**.** *We call $x \in \mathbb{R}^d$ an $\varepsilon$-accurate solution to problem* (1)*, if*

$$\|\nabla\Phi(x)\| \le \varepsilon.\tag{7}$$

It is worth remarking here that the inequality (7) can be modified to $\|\nabla\Phi(x)\| \le C\varepsilon$ for some *global* constant $C > 0$ without hurting the magnitude of IFO complexity bounds.

Let us first make some assumptions regarding to each component of the (compositional) objective function. Analogous to Assumption 1(i) in Fang et al. (2018), we make the following finite gap assumption:

**Assumption 1** (Finite gap)**.** *We assume that the algorithm is initialized at $x_0 \in \mathbb{R}^d$ with*

$$\Delta := \Phi(x_0) - \Phi^* < \infty,\tag{8}$$

*where $\Phi^*$ denotes the global minimum value of $\Phi(x)$.*

We make the following smoothness and boundedness assumptions, which are standard in recent compositional optimization literatures (e.g. Lian et al. (2017); Huo et al. (2018); Lin et al. (2018)).

**Assumption 2** (Smoothness)**.** *There exist Lipschitz constants $L_g, L_f, L_\Phi > 0$ such that for $i \in [1,n]$, $j \in [1,m]$ we have*

$$
\begin{array}{lll}
\|\partial g_j(x) - \partial g_j(x')\|_F & \le L_g\|x - x'\| & \text{for } x, x' \in \mathbb{R}^d,\\[4pt]
\|\nabla f_i(y) - \nabla f_i(y')\| & \le L_f\|y - y'\| & \text{for } y, y' \in \mathbb{R}^l,\\[4pt]
\left\|[\partial g_j(x)]^\top \nabla f_i(g(x)) - [\partial g_j(x')]^\top \nabla f_i(g(x'))\right\| & \le L_\Phi\|x - x'\| & \text{for } x, x' \in \mathbb{R}^d.
\end{array}\tag{9}
$$

---

**Algorithm 1** SARAH-Compositional, Online Case (resp. Finite-Sum Case)

---

**Input**: $T, q, x_0, \eta, S_1^L, S_2^L, S_3^L$
**for** $t = 0$ **to** $T - 1$ **do**
  **if** $\mod (t, q) = 0$ **then**
    Draw $S_1^L$ indices with replacement $\mathcal{S}_{1,t}^L \subseteq [1, m]$ and let $\boldsymbol{g}_t = \dfrac{1}{S_1^L} \sum\limits_{j \in \mathcal{S}_{1,t}^L} g_j(x_t)$

                                     (resp. $\boldsymbol{g}_t = g(x_t)$ in finite-sum case)

    Draw $S_2^L$ indices with replacement $\mathcal{S}_{2,t}^L \subseteq [1, m]$ and let $\boldsymbol{G}_t = \dfrac{1}{S_2^L} \sum\limits_{j \in \mathcal{S}_{2,t}^L} \partial g_j(x_t)$

                                       (resp. $\boldsymbol{G}_t = \partial g(x_t)$ in finite-sum case)

    Draw $S_3^L$ indices with replacement $\mathcal{S}_{3,t}^L \subseteq [1, n]$ and let $\boldsymbol{F}_t = (\boldsymbol{G}_t)^\top \left[ \dfrac{1}{S_3^L} \sum\limits_{i \in \mathcal{S}_{3,t}^L} \nabla f_i(\boldsymbol{g}_t) \right]$

                                    (resp. $\boldsymbol{F}_t = (\boldsymbol{G}_t)^\top \nabla f(\boldsymbol{g}_t)$ in finite-sum case)

  **else**
    Draw one index $j_t \in [1, m]$ and let $\boldsymbol{g}_t = g_{j_t}(x_t) - g_{j_t}(x_{t-1}) + \boldsymbol{g}_{t-1}$ and

$$\boldsymbol{G}_t = \partial g_{j_t}(x_t) - \partial g_{j_t}(x_{t-1}) + \boldsymbol{G}_{t-1}$$

    Draw one index $i_t \in [1, n]$ and let

$$\boldsymbol{F}_t = (\boldsymbol{G}_t)^\top \nabla f_{i_t}(\boldsymbol{g}_t) - (\boldsymbol{G}_{t-1})^\top \nabla f_{i_t}(\boldsymbol{g}_{t-1}) + \boldsymbol{F}_{t-1}$$

  **end if**
  Update $x_{t+1} = x_t - \eta \boldsymbol{F}_t$
**end for**
**return** Output $\widetilde{x}$ chosen uniformly at random from $\{x_t\}_{t=0}^{T-1}$

---

Here for the purpose of using stochastic recursive estimation of $\partial g(x)$, we slightly strengthen the smoothness assumption by adopting the Frobenius norm on the left hand of the first line of (9).

**Assumption 3** (Boundedness). *There exist boundedness constants $M_g, M_f > 0$ such that for $i \in [1, n]$, $j \in [1, m]$ we have*

$$
\begin{aligned}
\|\partial g_j(x)\| &\leq M_g \quad \text{for } x \in \mathbb{R}^d, \\
\|\nabla f_i(y)\| &\leq M_f \quad \text{for } y \in \mathbb{R}^l.
\end{aligned}
\tag{10}
$$

Notice that applying mean-value theorem for vector-valued functions to (10) gives another Lipschitz condition

$$\|g_j(x) - g_j(x')\| \leq M_g \|x - x'\| \qquad \text{for } x, x' \in \mathbb{R}^d, \tag{11}$$

and analogously for $f_i(y)$. It turns out that under the above two assumptions, a choice of $L_\Phi$ in (9) can be expressed as a polynomial of $L_f, L_g, M_f, M_g$. For clarity purposes in the rest of this paper, we adopt the following typical choice of $L_\Phi$

$$L_\Phi \equiv M_f L_g + M_g^2 L_f, \tag{12}$$

whose applicability can be verified via a simple application of the chain rule. We integrate both finite-sum and online cases into one algorithm SARAH-Compositional and write it in Algorithm 1.

## 3  Convergence Rate Analysis

In this section, we aim to justify that our proposed SARAH-Compositional algorithm provides IFO complexities of $\mathcal{O}((n + m)^{1/2} \varepsilon^{-2})$ in the finite-sum case and $\mathcal{O}(\varepsilon^{-3})$ in the online case, which supersedes the concurrent and comparative algorithms (see more in Table 1).

Let us first analyze the convergence in the finite-sum case. In this case we have $\mathcal{S}_1^L = [1, m]$, $\mathcal{S}_2^L = [1, m]$, $\mathcal{S}_3^L = [1, n]$. Involved analysis leads us to conclude

**Theorem 1** (Finite-sum case). *Suppose Assumptions 1, 2 and 3 in §2 hold, let $\mathcal{S}_1^L = \mathcal{S}_2^L = [1, m]$, $\mathcal{S}_3^L = [1, n]$, $q = (2m + n)/3$, and set the stepsize*

$$\eta = \frac{1}{\sqrt{6(2m + n)\left(M_g^4 L_f^2 + M_f^2 L_g^2\right)}}. \tag{13}$$

*Then for the finite-sum case, SARAH-Compositional Algorithm 1 outputs an $\widetilde{x}$ satisfying $\mathbf{E}\|\nabla\Phi(\widetilde{x})\|^2 \leq \varepsilon^2$ in*

$$\sqrt{2m + n} \cdot \sqrt{M_g^4 L_f^2 + M_f^2 L_g^2} \cdot \frac{\sqrt{24}[\Phi(x_0) - \Phi^*]}{\varepsilon^2} \tag{14}$$

*iterates. The IFO complexity to achieve an $\varepsilon$-accurate solution is bounded by*

$$2m + n + \sqrt{2m + n} \cdot \sqrt{M_g^4 L_f^2 + M_f^2 L_g^2} \cdot \frac{\sqrt{1944}[\Phi(x_0) - \Phi^*]}{\varepsilon^2}. \tag{15}$$

Theorem 1 allows us to achieve an $\varepsilon$-accurate solution, and a simple application of Markov's inequality allows us to derive high-probability results for achieving $\varepsilon$-accurate solutions. Compared with Fang et al. (2018), one observes that Theorem 1 indicates an IFO complexity upper bound of $\mathcal{O}(m + n + (m + n)^{1/2}\varepsilon^{-2})$ to achieve an $\varepsilon$-accurate solution, sharing a similar form with that of SARAH/SPIDER for non-convex stochastic optimization when $m + n$ is regarded as the number of individual functions.[4] SPIDER-SFO (as a SARAH variant) is *optimal* in both finite-sum and online cases, in the sense that it matches the theoretical lower bound (Fang et al., 2018; Arjevani et al., 2019), which makes it tempting to claim that our proposed SARAH-Compositional as its extension is also optimal. We emphasize that the set of assumptions for compositional optimization is different from vanilla optimization, and claiming optimality of the IFO complexity requires a corresponding lower bound result, left as a future direction to explore.

Let us then analyze the convergence in the online case, where we sample minibatches $\mathcal{S}_1^L, \mathcal{S}_2^L, \mathcal{S}_3^L$ of relevant quantities instead of the ground truth once every $q$ iterates. To characterize the estimation error, we put in one additional finite variance assumption:

**Assumption 4** (Finite Variance). *We assume that there exists $H_1$, $H_2$ and $H_3$ as the upper bounds on the variance of the functions $f(y)$, $\partial g(x)$, and $g(x)$, respectively, such that*

$$
\begin{aligned}
\mathbb{E}\|g_i(x) - g(x)\|^2 &\leq H_1 & \text{for } x \in \mathbb{R}^d, \\
\mathbb{E}\|\partial g_i(x) - \partial g(x)\|^2 &\leq H_2 & \text{for } x \in \mathbb{R}^d, \\
\mathbb{E}\|\nabla f_i(y) - \nabla f(y)\|^2 &\leq H_3 & \text{for } y \in \mathbb{R}^l.
\end{aligned}
\tag{16}
$$

From Assumptions 2 and 3 we can easily verify, via triangle inequality and convexity of norm, that $H_2$ can be chosen as $4M_g^2$ and $H_3$ can be chosen as $4M_f^2$. On the contrary, $H_1$ *cannot* be represented as a function of boundedness and smoothness constants. We conclude the following theorem for the online case:

**Theorem 2** (Online case). *Suppose Assumptions 1, 2, 3 and 4 in §2 hold, let $S_1^L = \dfrac{3H_1 M_g^2 L_f^2}{\varepsilon^2}$, $S_2^L = \dfrac{3H_2 M_f^2}{\varepsilon^2}$, $S_3^L = \dfrac{3H_3 M_g^2}{\varepsilon^2}$, let $q = \dfrac{D_0}{3\varepsilon^2}$ where we denote the noise-relevant parameter*

$$D_0 := 3\left(H_1 M_g^2 L_f^2 + H_2 M_f^2 + H_3 M_g^2\right), \tag{17}$$

*and set the stepsize*

$$\eta = \frac{\varepsilon}{\sqrt{6D_0\left(M_g^4 L_f^2 + M_f^2 L_g^2\right)}}. \tag{18}$$

*Then for the online case, SARAH-Compositional Algorithm 1 outputs an $\widetilde{x}$ satisfying $\mathbf{E}\|\nabla\Phi(\widetilde{x})\|^2 \leq 2\varepsilon^2$ in*

$$\sqrt{D_0} \cdot \sqrt{M_g^4 L_f^2 + M_f^2 L_g^2} \cdot \frac{\sqrt{24}[\Phi(x_0) - \Phi^*]}{\varepsilon^3} \tag{19}$$

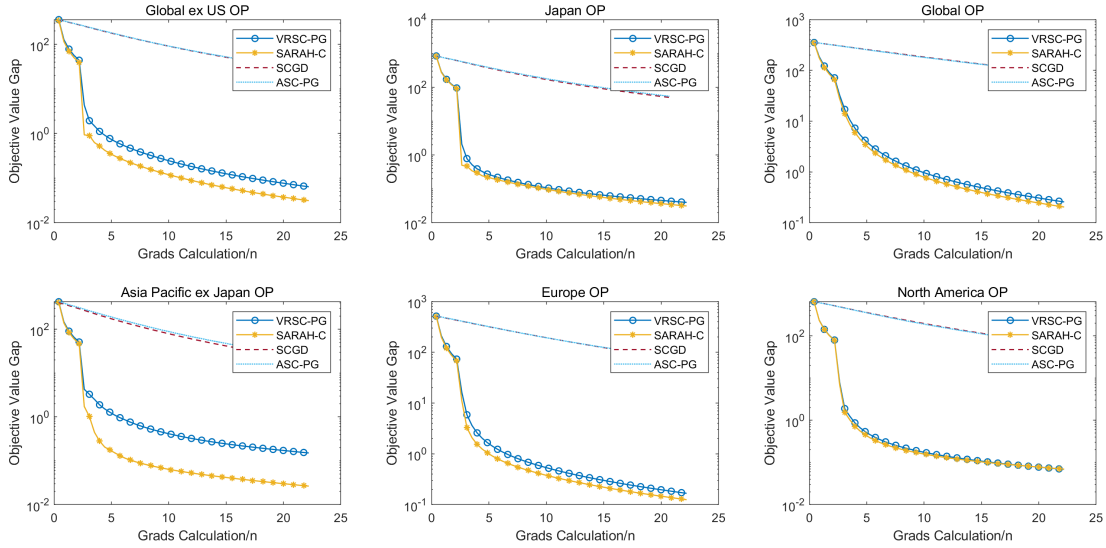

Figure 1: Experiment on the portfolio management. The $x$-axis is the number of gradients calculations divided by the number of samples, the $y$-axis is the function value gap.

*iterates. The IFO complexity to achieve an $\varepsilon$-accurate solution is bounded by*

$$\frac{D_0}{\varepsilon^2} + \sqrt{D_0} \cdot \sqrt{M_g^4 L_f^2 + M_f^2 L_g^2} \cdot \frac{\sqrt{1944}[\Phi(x_0) - \Phi^*]}{\varepsilon^3}. \tag{20}$$

We see that in the online case, the IFO complexity to achieve an $\varepsilon$-accurate solution is upper bounded by $\mathcal{O}(\varepsilon^{-3})$. Due to space limits, the detailed proofs of Theorems 1 and 2 are deferred to the supplementary material.

## 4 Experiments

In this section, we study performance of our algorithm to risk-adverse portfolio management problem and conduct numerical experiments to support our theory.[5] We follow the setups in Huo et al. (2018); Liu et al. (2017) and compare with existing algorithms for compositional optimization. Readers are referred to Wang et al. (2017a) for more tasks our algorithm can be potentially applied for.

Recall that in §1, we formulate our portfolio management problem as a mean-variance optimization problem (2), which can be formulated as a compositional optimization problem (1). As it satisfies Assumptions 1–4 in a bounded domain of optimization, it serves as a good example to validate our theory. For convenience we repeat the display here:

$$\min_{x \in \mathbb{R}^N} \quad -\frac{1}{T}\sum_{t=1}^{T}\langle r_t, x\rangle + \frac{1}{T}\sum_{t=1}^{T}\left(\langle r_t, x\rangle - \frac{1}{T}\sum_{s=1}^{T}\langle r_s, x\rangle\right)^2, \tag{2}$$

where $x = \{x_1, x_2, \ldots, x_N\} \in \mathbb{R}^N$ denotes the quantities invested at every asset $i = 1, \ldots, N$.

When applying SARAH-Compositional we adopt the online case where we pick $S_1^L, S_2^L, S_3^L$ as the mini-batch sizes once every $q$ steps. Datasets include different portfolio datas formed on Size and Operating Profitability.[6] We choose to use 6 different 25-portfolio datasets where $N = 25$ and $T = 7240$, same as the ones adopted by Lin et al. (2018). Specifically, we choose $S_1^L = S_2^L = S_3^L = 2000$ (roughly optimized to improve the numerical performance). The results are shown in Figure 1.

We demonstrate the comparison among our algorithm SARAH-Compositional, SCGD (Wang et al., 2017a), ASC-PG (Wang et al., 2017b) and VRSC-PG (Huo et al., 2018) (serving as a baseline for variance-reduced stochastic compositional optimization methods). We plot the objective function value gap and gradient norm against IFO complexity (measured by gradients calculation) for all four algorithms in two covariance settings and six real-world datasets. We observe that SARAH-Compositional outperforms all comparable algorithms. Our range of stepsize is $\left\{1 \times 10^{-5}, 1 \times 10^{-4}, 2 \times 10^{-4}, 5 \times 10^{-4}, 1 \times 10^{-3}, 1 \times 10^{-2}\right\}$, and we plot the learning curve for each algorithm corresponding to their individually optimized stepsize. For SCGD and ASC-PG algorithms, we fix the extrapolation parameter $\beta$ as 0.9. The $q$-parameters in both SARAH-Compositional and VRSC-PG algorithms are set as 50.

The toy experiment provides evidence that our proposed SARAH-Compositional algorithm applied to risk-adverse portfolio management problem achieves state-of-the art performance. Moreover, we note that due to the small mini-batch sizes, basic SCGD achieves a less satisfactory result, a phenomenon also shown by Huo et al. (2018); Lian et al. (2017).

## 5   Conclusion

In this paper, we propose a novel algorithm called SARAH-Compositional for solving stochastic compositional optimization problems using the idea of a recently proposed variance reduced gradient method. Our algorithm achieves both outstanding theoretical and experimental results. Theoretically, we show that the SARAH-Compositional algorithm can achieve desirable efficiency and IFO upper bound complexities for finding an $\varepsilon$-accurate solution of non-convex compositional problems in both finite-sum and online cases. Theoretically, we show that the SARAH-Compositional algorithm can achieve improved convergence rates and IFO complexities for finding an $\varepsilon$-accurate solution to non-convex compositional problems in both finite-sum and online cases. Experimentally, we compare our new compositional optimization method with a few rival algorithms for the task of portfolio management and demonstrate its superior performance. Future directions include handling the non-smooth case and the theory of lower bounds for stochastic compositional optimization. We hope this work can provide new perspectives to both optimization and machine learning communities interested in compositional optimization.

## Footnotes

[2]This is also referred to as stochastic recursive variance reduction method, incremental variance reduction method or SPIDER-BOOST in various recent literatures. We stick to name the algorithm after SARAH to respect to our best knowledge the earliest discovery of that algorithm.

[3]Wang et al. (2019) names their algorithm SPIDER-BOOST since it can be seen as the SPIDER-SFO algorithm with relaxed step-length restrictions.

[4]Here and in below, the smoothness and boundedness parameters and $\Phi(x_0) - \Phi^*$ are treated as constants.

[5]The source code can be found at `http://github.com/angeoz/SCGD`. Space limiting, we refer the readers to the full version of this paper for the experiment studies of other applications including reinforcement learning and stochastic neighborhood embedding.

[6]`http://mba.tuck.dartmouth.edu/pages/faculty/ken.french/data_library.html`

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
