[Supplementary Material]

# A Detailed Analysis of Convergence Theorems

In this section, we detail the analysis of our Theorems 1 and 2. Before moving on, we first provide a key lemma that serves as their common analysis, whose proof is provided in §A.3. We assume that the expected estimation error squared is bounded as the following for any $t$ and some parameters $\omega_1$, $\omega_2$ and $\omega_3$ to be specified here:

**Lemma 1.** *Assume that for any initial point $x_0 \in \mathbb{R}^d$*

$$\mathbb{E}\|\boldsymbol{g}_0 - g(x_0)\|^2 = \mathbb{E}\left\|\frac{1}{S_1^L} \sum_{j \in \mathcal{S}_{1,0}^L} g_j(x_0) - g(x_0)\right\|^2 \leq \omega_1,$$

$$\mathbb{E}\|\boldsymbol{G}_0 - \partial g(x_0)\|^2 = \mathbb{E}\left\|\frac{1}{S_2^L} \sum_{j \in \mathcal{S}_{2,0}^L} \partial g_j(x_0) - \partial g(x_0)\right\|^2 \leq \omega_2, \tag{21}$$

$$\mathbb{E}\|\boldsymbol{F}_0 - (\boldsymbol{G}_0)^\top \nabla f(\boldsymbol{g}_0)\|^2 = \mathbb{E}\left\|(\boldsymbol{G}_0)^\top \left[\frac{1}{S_3^L} \sum_{i \in \mathcal{S}_{3,0}^L} \nabla f_i(\boldsymbol{g}_0)\right] - (\boldsymbol{G}_0)^\top \nabla f(\boldsymbol{g}_0)\right\|^2 \leq \omega_3.$$

*Then we have*

$$\mathbb{E}\|\nabla\Phi(\widetilde{x})\|^2 \leq \frac{2}{T\eta}[\Phi(x_0) - \Phi^*] + 3\left(M_g^2 L_f^2 \omega_1 + M_f^2 \omega_2 + \omega_3\right). \tag{22}$$

With Lemma 1, our goal is to make the left hand of (22) no greater than $\mathcal{O}(\varepsilon^2)$. We present the proofs for finite-sum and online cases, separately.

## A.1 Proof of Theorem 1

*Proof of Theorem 1.* In this finite-sum case for Algorithm 1, $\omega_1 = \omega_2 = \omega_3 = 0$. Bringing into (22), in order to achieve $\leq \varepsilon^2$ for $\mathbb{E}[\|\nabla\Phi(\widetilde{x})\|^2]$ we need $\frac{2}{T\eta}(\Phi(x_0) - \Phi^*) \leq \varepsilon^2$. Recalling the choice of stepsize in (13), the total iteration complexity $Q_{\text{iter}}$ is

$$Q_{\text{iter}} = \frac{2[\Phi(x_0) - \Phi^*]}{\varepsilon^2} \cdot \eta^{-1} = \frac{2[\Phi(x_0) - \Phi^*]}{\varepsilon^2} \cdot \sqrt{6(2m+n)\left(M_g^4 L_f^2 + M_f^2 L_g^2\right)}, \tag{23}$$

proving (14). Note that $S_1^L = m = S_2^L, S_3^L = n$, the total IFO complexity achieving $\varepsilon$-accurate solution is hence bounded by

$$\text{IFO complexity} = (S_1^L + S_2^L + S_3^L)\left\lceil\frac{Q_{\text{iter}}}{q}\right\rceil + 6\left(Q_{\text{iter}} - \left\lceil\frac{Q_{\text{iter}}}{q}\right\rceil\right)$$

$$\leq 2m + n + \left(\frac{2m+n}{q} + 6\right) \cdot \frac{2[\Phi(x_0) - \Phi^*]}{\varepsilon^2} \cdot \sqrt{6(2m+n)\left(M_g^4 L_f^2 + M_f^2 L_g^2\right)}$$

$$\leq 2m + n + \sqrt{2m+n} \cdot \sqrt{M_g^4 L_f^2 + M_f^2 L_g^2} \cdot \frac{\sqrt{1944}[\Phi(x_0) - \Phi^*]}{\varepsilon^2}.$$

This completes our proof of (15) and the whole theorem.

$\square$

## A.2 Proof of Theorem 2

*Proof of Theorem 2.* In the online case for Algorithm 1, from (22) we need both $\frac{2}{T\eta}(\Phi(x_0) - \Phi^*) \leq \varepsilon^2$ and $3(M_g^2 L_f^2 \omega_1 + M_f^2 \omega_2 + \omega_3) \leq \varepsilon^2$ to achieve $\mathbb{E}[\|\nabla\Phi(\tilde{x})\|^2] \leq 2\varepsilon^2$ (factor 2 here is for consistency of parameter choice). Recalling (28), the total iteration complexity $Q_{\text{iter}}$ is

$$Q_{\text{iter}} = \frac{2[\Phi(x_0) - \Phi^*]}{\varepsilon^2} \cdot \eta^{-1} = \frac{2[\Phi(x_0) - \Phi^*]}{\varepsilon^2} \cdot \sqrt{6\left(M_g^4 L_f^2 + M_f^2 L_g^2\right)\frac{D_0}{\varepsilon^2}}, \tag{24}$$

proving (19). Note that $S_1^L = \dfrac{3H_1 M_g^2 L_f^2}{\varepsilon^2}, S_2^L = \dfrac{3H_2 M_f^2}{\varepsilon^2}, S_3^L = \dfrac{3H_3 M_g^2}{\varepsilon^2}$, we derive from (17) that $S_1^L + S_2^L + S_3^L = \dfrac{D_0}{\varepsilon^2}$ and $q = \dfrac{D_0}{3\varepsilon^2}$, and the total IFO complexity achieving $\varepsilon$-accurate solution is hence bounded by

$$
\begin{aligned}
\text{IFO complexity} &= (S_1^L + S_2^L + S_3^L) \left\lceil \frac{Q_{\text{iter}}}{q} \right\rceil + 6 \left( Q_{\text{iter}} - \left\lceil \frac{Q_{\text{iter}}}{q} \right\rceil \right) \\
&\leq \frac{D_0}{\varepsilon^2} + \left( \frac{D_0/\varepsilon^2}{D_0/(3\varepsilon^2)} + 6 \right) Q_{\text{iter}} \\
&\leq \frac{D_0}{\varepsilon^2} + 9 \cdot \frac{2[\Phi(x_0) - \Phi^*]}{\varepsilon^2} \cdot \sqrt{6 \left( M_g^4 L_f^2 + M_f^2 L_g^2 \right) \frac{D_0}{\varepsilon^2}} \\
&= \frac{D_0}{\varepsilon^2} + \sqrt{D_0} \cdot \sqrt{M_g^4 L_f^2 + M_f^2 L_g^2} \cdot \frac{\sqrt{1944}[\Phi(x_0) - \Phi^*]}{\varepsilon^3}.
\end{aligned}
$$

This completes our proof of (20) and the whole theorem.

$\square$

## A.3 Proof of Lemma 1

Before starting our proof, we first prove the following three lemmas that characterize the error bounds induced by our iterations. For approximation errors for $g$-iteration and $G$-iteration, we prove the following

**Lemma 2** (Error bound induced by $g$-iteration). *We have for any fixed $t \geq 0$*

$$
\mathbb{E} \left\| \nabla \Phi\left(x_t\right) - \left(\partial g\left(x_t\right)\right)^\top \nabla f\left(\boldsymbol{g}_t\right) \right\|^2 \leq M_g^2 L_f^2 \omega_1 + M_g^4 L_f^2 \cdot \eta^2 \sum_{s=\lfloor t/q \rfloor q + 1}^t \mathbb{E} \|\boldsymbol{F}_{s-1}\|^2, \quad (25)
$$

*where the expectation in the last term is taken over a uniformly chosen $j$ in $[1, m]$.*

In addition, we have

**Lemma 3** (Error bound induced by $G$-iteration). *We have for any fixed $t \geq 0$*

$$
\mathbb{E} \left\| \left(\partial g(x_t)\right)^\top \nabla f(\boldsymbol{g}_t) - \left(\boldsymbol{G}_t\right)^\top \nabla f(\boldsymbol{g}_t) \right\|^2 \leq M_f^2 \omega_2 + M_f^2 L_g^2 \cdot \eta^2 \sum_{s=\lfloor t/q \rfloor q + 1}^t \mathbb{E} \|\boldsymbol{F}_{s-1}\|^2, \quad (26)
$$

*where the expectation in the last term is taken over a uniformly chosen $j$ in $[1, m]$.*

Finally, we prove the following lemma that characterizes the gap between $\boldsymbol{F}$-iteration and $\left(\boldsymbol{G}_t\right)^\top \nabla f\left(\boldsymbol{g}_t\right)$:

**Lemma 4** (Error bound induced by $\boldsymbol{F}$-iteration). *We have for any fixed $t \geq 0$*

$$
\mathbb{E} \left\| \left(\boldsymbol{G}_t\right)^\top \nabla f\left(\boldsymbol{g}_t\right) - \boldsymbol{F}_t \right\|^2 \leq \omega_3 + 2 \left( M_g^4 L_f^2 + M_f^2 L_g^2 \right) \cdot \eta^2 \sum_{s=\lfloor t/q \rfloor q + 1}^t \mathbb{E} \|\boldsymbol{F}_{s-1}\|^2, \quad (27)
$$

*where the expectation in the last term is taken over a uniformly chosen $i$ in $[1, n]$.*

*Proof of Lemma 1.* We prove the lemma in the following steps:

(i) Assume the settings in Theorem 1 hold. We come to show, from (12) and (13), that

$$
\frac{M_f L_g + M_g^2 L_f}{\sqrt{6(2m+n) \left( M_g^4 L_f^2 + M_f^2 L_g^2 \right)}} \equiv L_\Phi \eta \leq \frac{1}{2}. \quad (28)
$$

This is equivalent to showing

$$
2 \left( M_f L_g + M_g^2 L_f \right)^2 \leq 3(2m+n) \left( M_g^4 L_f^2 + M_f^2 L_g^2 \right).
$$

This is concluded naturally from $4 < 9 \leq 3(2m + n)$ and hence

$$2 \left( M_f L_g + M_g^2 L_f \right)^2 \leq 4 \left( M_f^2 L_g^2 + M_g^4 L_f^2 \right) \leq 3(2m + n) \left( M_g^4 L_f^2 + M_f^2 L_g^2 \right).$$

(ii) Standard arguments along with the smoothness Assumption 2, we have from the update rule is $x_{t+1} = x_t - \eta \boldsymbol{F}_t$ and (9) that for any $x, x' \in \mathbb{R}^d$,

$$\|\nabla \Phi(x) - \nabla \Phi(x')\| = \left\| [\partial g(x)]^\top \nabla f(g(x)) - [\partial g(x')]^\top \nabla f(g(x')) \right\|$$

$$\leq \frac{1}{nm} \sum_{j=1}^m \sum_{i=1}^n \left\| [\partial g_j(x)]^\top \nabla f_i(g(x)) - [\partial g_j(x')]^\top \nabla f_i(g(x')) \right\|$$

$$\leq L_\Phi \|x - x'\|,$$

and hence a Taylor's expansion argument gives

$$\Phi(x_{t+1}) \leq \Phi(x_t) + (\nabla \Phi(x_t))^\top (x_{t+1} - x_t) + \frac{L_\Phi}{2} \|x_{t+1} - x_t\|^2$$

$$= \Phi(x_t) - \frac{\eta}{2} \cdot 2(\nabla \Phi(x_t))^\top \boldsymbol{F}_t + \frac{L_\Phi \eta^2}{2} \|\boldsymbol{F}_t\|^2$$

$$\leq \Phi(x_t) - \frac{\eta}{2} \|\nabla \Phi(x_t)\|^2 + \frac{\eta}{2} \|\nabla \Phi(x_t) - \boldsymbol{F}_t\|^2 - \left( \frac{\eta}{2} - \frac{L_\Phi \eta^2}{2} \right) \|\boldsymbol{F}_t\|^2,$$

$$\leq \Phi(x_t) - \frac{\eta}{2} \|\nabla \Phi(x_t)\|^2 + \frac{\eta}{2} \left( \|\nabla \Phi(x_t) - \boldsymbol{F}_t\|^2 - \frac{1}{2} \|\boldsymbol{F}_t\|^2 \right),$$

where the second to last inequality above follows from (13) and the fact $2a^\top b = \|a\|^2 - \|a - b\|^2 + \|b\|^2$ for any real vectors $a, b$, and the last inequality is due to $L_\Phi \eta \leq 1/2$ given by (28). Summing the above over $t = 0, \ldots, T - 1$ and taking expectation on both sides allow us to conclude

$$\Phi^* \leq \mathbb{E}\Phi(x_T) \leq \Phi(x_0) - \frac{\eta}{2} \sum_{t=0}^{T-1} \mathbb{E}\|\nabla \Phi(x_t)\|^2$$

$$+ \frac{\eta}{2} \left( \sum_{t=0}^{T-1} \mathbb{E}\|\nabla \Phi(x_t) - \boldsymbol{F}_t\|^2 - \frac{1}{2} \sum_{t=0}^{T-1} \mathbb{E}\|\boldsymbol{F}_t\|^2 \right). \tag{29}$$

Since $\widetilde{x}$ is chosen uniformly at random from $\{x_t\}_{t=0}^{T-1}$, rearranging (29) gives

$$\mathbb{E}\|\nabla \Phi(\widetilde{x})\|^2 = \frac{1}{T} \sum_{t=0}^{T-1} \mathbb{E}\|\nabla \Phi(x_t)\|^2$$

$$\leq \frac{2}{T\eta} [\Phi(x_0) - \Phi^*] + \frac{1}{T} \left( \sum_{t=0}^{T-1} \mathbb{E}\|\nabla \Phi(x_t) - \boldsymbol{F}_t\|^2 - \frac{1}{2} \sum_{t=0}^{T-1} \mathbb{E}\|\boldsymbol{F}_t\|^2 \right). \tag{30}$$

(iii) To bound $\|\nabla \Phi(x_t) - \boldsymbol{F}_t\|^2$ in expectation, note

$$\mathbb{E}\|\nabla \Phi(x_t) - \boldsymbol{F}_t\|^2$$

$$= \mathbb{E}\left\| \nabla \Phi(x_t) - (\partial g(x_t))^\top \nabla f(\boldsymbol{g}_t) - (\partial g(x_t))^\top \nabla f(\boldsymbol{g}_t) + (\boldsymbol{G}_t)^\top \nabla f(\boldsymbol{g}_t) - (\boldsymbol{G}_t)^\top \nabla f(\boldsymbol{g}_t) + \boldsymbol{F}_t \right\|^2$$

$$\leq 3\mathbb{E}\left\| \nabla \Phi(x_t) - (\partial g(x_t))^\top \nabla f(\boldsymbol{g}_t) \right\|^2 + 3\mathbb{E}\left\| (\partial g(x_t))^\top \nabla f(\boldsymbol{g}_t) - (\boldsymbol{G}_t)^\top \nabla f(\boldsymbol{g}_t) \right\|^2$$

$$+ 3\mathbb{E}\left\| (\boldsymbol{G}_t)^\top \nabla f(\boldsymbol{g}_t) - \boldsymbol{F}_t \right\|^2, \tag{31}$$

where in the last inequality we applied Minkowski's inequality (along with elementary algebra).

The three terms in (31) can be estimated using a combination of Lemmas 2, 3 and 4, where we have

$$\mathbb{E}\|\nabla \Phi(x_t) - \boldsymbol{F}_t\|^2 \leq 3 \left( M_g^2 L_f^2 \omega_1 + M_f^2 \omega_2 + \omega_3 \right) + 9 \left( M_g^4 L_f^2 + M_f^2 L_g^2 \right) \eta^2 \sum_{s=\lfloor t/q \rfloor q + 1}^{t} \mathbb{E}\|\boldsymbol{F}_{s-1}\|^2. \tag{32}$$

Back to (30) which is repeated in below

$$\mathbb{E}\|\nabla\Phi(\widetilde{x})\|^2 = \frac{1}{T}\sum_{t=0}^{T-1}\mathbb{E}\|\nabla\Phi(x_t)\|^2$$

$$\leq \frac{2}{T\eta}[\Phi(x_0) - \Phi^*] + \frac{1}{T}\left(\sum_{t=0}^{T-1}\mathbb{E}\|\nabla\Phi(x_t) - \boldsymbol{F}_t\|^2 - \frac{1}{2}\sum_{t=0}^{T-1}\mathbb{E}\|\boldsymbol{F}_t\|^2\right). \qquad (30)$$

For the second term of (30), note that aggregating (32) for each $q$-step epoch implies, when stepsize $\eta$ picked as in (13) satisfies $9\left(M_g^4 L_f^2 + M_f^2 L_g^2\right)q\eta^2 - 1/2 \leq 0$, the following

$$\frac{1}{T}\left(\sum_{t=0}^{T-1}\mathbb{E}\|\nabla\Phi(x_t) - \boldsymbol{F}_t\|^2 - \frac{1}{2}\sum_{t=0}^{T-1}\mathbb{E}\|\boldsymbol{F}_t\|^2\right)$$

$$\leq \frac{1}{T}\sum_{t=0}^{T-1}\left(3\left(M_g^2 L_f^2\omega_1 + M_f^2\omega_2 + \omega_3\right) + 9\left(M_g^4 L_f^2 + M_f^2 L_g^2\right)\eta^2\sum_{s=\lfloor t/q\rfloor q+1}^{t}\mathbb{E}\|\boldsymbol{F}_{s-1}\|^2\right)$$

$$-\frac{1}{2}\cdot\frac{1}{T}\sum_{t=0}^{T-1}\mathbb{E}\|\boldsymbol{F}_t\|^2$$

$$\overset{(32)}{\leq} \frac{1}{T}\cdot T\cdot 3(M_g^2 L_f^2\omega_1 + M_f^2\omega_2 + \omega_3) + 9\left(M_g^4 L_f^2 + M_f^2 L_g^2\right)q\eta^2\cdot\frac{1}{T}\sum_{t=0}^{T-1}\mathbb{E}\|\boldsymbol{F}_t\|^2$$

$$-\frac{1}{2}\cdot\frac{1}{T}\sum_{t=0}^{T-1}\mathbb{E}\|\boldsymbol{F}_t\|^2$$

$$= 3(M_g^2 L_f^2\omega_1 + M_f^2\omega_2 + \omega_3) + \left(9\left(M_g^4 L_f^2 + M_f^2 L_g^2\right)q\eta^2 - \frac{1}{2}\right)\cdot\frac{1}{T}\sum_{t=0}^{T-1}\mathbb{E}\|\boldsymbol{F}_t\|^2$$

$$\leq 3(M_g^2 L_f^2\omega_1 + M_f^2\omega_2 + \omega_3). \qquad (33)$$

(30) and (33) together conclude (22) and hence Lemma 1.

$$\square$$

# B  Detailed Proofs of Auxiliary Lemmas

## B.1  Proof of Lemma 2

*Proof of Lemma 2.* We prove the lemma for the case of $t < q$, and for other $t$ it applies directly due to Markov property when epochs (as vectors) are viewed as states of the Markov chain.

(i) We first bound the $\mathbb{E}\|\boldsymbol{g}_t - g(x_t)\|^2$ term and show that for any fixed $t \geq 0$

$$\mathbb{E}\|\boldsymbol{g}_t - g(x_t)\|^2 \leq \mathbb{E}\|\boldsymbol{g}_{\lfloor t/q\rfloor q} - g(x_{\lfloor t/q\rfloor q})\|^2 + \sum_{s=\lfloor t/q\rfloor q+1}^{t}\mathbb{E}\|g_{j_t}(x_s) - g_{j_t}(x_{s-1})\|^2. \quad (34)$$

First, we take expectation with respect to $j_t$ and get $\mathbb{E}g_{j_t}(x_t) = g(x_t)$ and have

$$\mathbb{E}\|\boldsymbol{g}_t - g(x_t)\|^2 = \mathbb{E}\|\boldsymbol{g}_{t-1} + g_{j_t}(x_t) - g_{j_t}(x_{t-1}) - g(x_t)\|^2$$

$$= \mathbb{E}\|\boldsymbol{g}_{t-1} - g(x_{t-1}) + g(x_{t-1}) + g_{j_t}(x_t) - g_{j_t}(x_{t-1}) - g(x_t)\|^2$$

$$= \mathbb{E}\|\boldsymbol{g}_{t-1} - g(x_{t-1})\|^2 + \mathbb{E}\|g_{j_t}(x_t) - g_{j_t}(x_{t-1}) - (g(x_t) - g(x_{t-1}))\|^2$$

$$\leq \mathbb{E}\|\boldsymbol{g}_{t-1} - g(x_{t-1})\|^2 + \mathbb{E}\|g_{j_t}(x_t) - g_{j_t}(x_{t-1})\|^2,$$

where we used $\mathbb{E}\|\boldsymbol{X} - \mathbb{E}[\boldsymbol{X}|\mathcal{F}]\|^2 \leq \mathbb{E}\|\boldsymbol{X}\|^2$ for any random vector $\boldsymbol{X}$ and any conditional expectation $\mathbb{E}[\boldsymbol{X}|\mathcal{F}]$. Apply the above calculations recursively proves (34).

(ii) We have for the left hand of (25)

$$\mathbb{E}\left\|\nabla\Phi\left(x_t\right) - \left(\partial g\left(x_t\right)\right)^\top \nabla f\left(\boldsymbol{g}_t\right)\right\|^2 = \mathbb{E}\left\|\left(\partial g\left(x_t\right)\right)^\top \nabla f\left(g(x_t)\right) - \left(\partial g\left(x_t\right)\right)^\top \nabla f\left(\boldsymbol{g}_t\right)\right\|^2$$
$$\leq M_g^2 L_f^2 \mathbb{E}\|\boldsymbol{g}_t - g(x_t)\|^2. \tag{35}$$

Applying (34) we obtain

$$\mathbb{E}\|\boldsymbol{g}_t - g(x_t)\|^2 \leq \mathbb{E}\|\boldsymbol{g}_{\lfloor t/q\rfloor q} - g(x_{\lfloor t/q\rfloor q})\|^2 + \sum_{s=1}^{t} \mathbb{E}\|g_{j_t}\left(x_s\right) - g_{j_t}\left(x_{s-1}\right)\|^2 \tag{36}$$
$$\leq \omega_1 + M_g^2 \cdot \eta^2 \sum_{s=1}^{t} \mathbb{E}\|\boldsymbol{F}_{s-1}\|^2.$$

Combining (36) and (35) together concludes the proof.

$\square$

## B.2  Proof of Lemma 3

*Proof of Lemma 3.* Analogous to Lemma 2 we only consider the case $t < q$.

(i) To begin with, we bound the $\mathbb{E}\left\|\boldsymbol{G}_t - \partial g(x_t)\right\|_F^2$ term (note the Frobenius norm) and conclude

$$\mathbb{E}\|\boldsymbol{G}_t - \partial g(x_t)\|_F^2 \leq \mathbb{E}\|\boldsymbol{G}_{\lfloor t/q\rfloor q} - \partial g(x_{\lfloor t/q\rfloor q})\|_F^2 + \sum_{s=\lfloor t/q\rfloor q+1}^{t} \mathbb{E}\|\partial g_{j_t}\left(x_s\right) - \partial g_{j_t}\left(x_{s-1}\right)\|_F^2. \tag{37}$$

In fact, following the techniques in the proof of (34) we have

$$\mathbb{E}\|\boldsymbol{G}_t - \partial g(x_t)\|_F^2 = \mathbb{E}\|\boldsymbol{G}_{t-1} + \partial g_{j_t}\left(x_t\right) - \partial g_{j_t}\left(x_{t-1}\right) - \partial g(x_t)\|_F^2$$
$$= \mathbb{E}\|\boldsymbol{G}_{t-1} - \partial g(x_{t-1}) + \partial g(x_{t-1}) + \partial g_{j_t}\left(x_t\right) - \partial g_{j_t}\left(x_{t-1}\right) - \partial g(x_t)\|_F^2$$
$$= \mathbb{E}\|\boldsymbol{G}_{t-1} - \partial g(x_{t-1})\|_F^2 + \mathbb{E}\|\partial g_{j_t}\left(x_t\right) - \partial g_{j_t}\left(x_{t-1}\right) - \left(\partial g(x_t) - \partial g(x_{t-1})\right)\|_F^2$$
$$\leq \mathbb{E}\|\boldsymbol{G}_{t-1} - \partial g(x_{t-1})\|_F^2 + \mathbb{E}\|\partial g_{j_t}\left(x_t\right) - \partial g_{j_t}\left(x_{t-1}\right)\|_F^2.$$

Recursively applying the above gives (37).

(ii) Further, note that for any fixed $t \geq 0$

$$\mathbb{E}\left\|\left(\partial g(x_t)\right)^\top \nabla f(\boldsymbol{g}_t) - \left(\boldsymbol{G}_t\right)^\top \nabla f(\boldsymbol{g}_t)\right\|^2 \leq M_f^2 \mathbb{E}\left\|\boldsymbol{G}_t - \partial g(x_t)\right\|^2. \tag{38}$$

Applying (37) we obtain, by smoothness condition (9), that

$$\mathbb{E}\|\boldsymbol{G}_t - \partial g(x_t)\|_F^2 \leq \mathbb{E}\|\boldsymbol{G}_{\lfloor t/q\rfloor q} - \partial g(x_{\lfloor t/q\rfloor q})\|_F^2 + \sum_{s=\lfloor t/q\rfloor q+1}^{t} \mathbb{E}\|\partial g_{j_t}\left(x_s\right) - \partial g_{j_t}\left(x_{s-1}\right)\|_F^2$$

$$\leq \omega_2 + L_g^2 \cdot \eta^2 \sum_{s=1}^{t} \mathbb{E}\|\boldsymbol{F}_{s-1}\|_F^2.$$

Bringing this into (38) and note the relation $\|\cdot\| \leq \|\bullet\|_F$ for a real matrix we conclude (26).

$\square$

## B.3  Proof of Lemma 4

*Proof of Lemma 4.* We prove this for $t < q$, and for other $t$ it follows the same procedure to prove. This prove is essentially the same reasoning as (34) and (37), but is significantly more lengthy due to handling more terms.

(i) First of all, we conclude that for any fixed $t \geq 0$

$$\mathbb{E}\left\|\boldsymbol{F}_t - (\boldsymbol{G}_t)^\top \nabla f(\boldsymbol{g}_t)\right\|^2 \leq \mathbb{E}\left\|\boldsymbol{F}_{\lfloor t/q\rfloor q} - (\boldsymbol{G}_{\lfloor t/q\rfloor q})^\top \nabla f(\boldsymbol{g}_{\lfloor t/q\rfloor q})\right\|^2$$
$$+ \sum_{s=1}^{t} \mathbb{E}\left\|(\boldsymbol{G}_s)^\top \nabla f_{i_t}(\boldsymbol{g}_s) - (\boldsymbol{G}_{s-1})^\top \nabla f_{i_t}(\boldsymbol{g}_{s-1})\right\|^2. \tag{39}$$

We unfold $\boldsymbol{F}_t$ using the update rule to get

$$\mathbb{E}\left\|\boldsymbol{F}_t - (\boldsymbol{G}_t)^\top \nabla f(\boldsymbol{g}_t)\right\|^2$$
$$= \mathbb{E}\left\|\boldsymbol{F}_{t-1} - (\boldsymbol{G}_{t-1})^\top \nabla f_{i_t}(\boldsymbol{g}_{t-1}) + (\boldsymbol{G}_t)^\top \nabla f_{i_t}(\boldsymbol{g}_t) - (\boldsymbol{G}_t)^\top \nabla f(\boldsymbol{g}_t)\right\|^2. \tag{40}$$

Subtracting and adding an auxiliary term $(\boldsymbol{G}_t)^\top \nabla f(\boldsymbol{g}_t)$, we result in an equivalent expression with the RHS of (40) being

$$\mathbb{E}\left\|\boldsymbol{F}_{t-1} - (\boldsymbol{G}_{t-1})^\top \nabla f(\boldsymbol{g}_{t-1})\right.$$
$$\left. + (\boldsymbol{G}_{t-1})^\top \nabla f(\boldsymbol{g}_{t-1}) - (\boldsymbol{G}_{t-1})^\top \nabla f_{i_t}(\boldsymbol{g}_{t-1}) + (\boldsymbol{G}_t)^\top \nabla f_{i_t}(\boldsymbol{g}_t) - (\boldsymbol{G}_t)^\top \nabla f(\boldsymbol{g}_t)\right\|^2.$$

We note that $\mathbb{E}\left[(\boldsymbol{G}_{t-1})^\top \nabla f(\boldsymbol{g}_{t-1}) - (\boldsymbol{G}_{t-1})^\top \nabla f_{i_t}(\boldsymbol{g}_{t-1}) + (\boldsymbol{G}_t)^\top \nabla f_{i_t}(\boldsymbol{g}_t) - (\boldsymbol{G}_t)^\top \nabla f(\boldsymbol{g}_t)\right] = 0$. Taking expectation with respect to $i_t$ before taking total expectation we result in a recursion:

$$\mathbb{E}\left\|\boldsymbol{F}_t - (\boldsymbol{G}_t)^\top \nabla f(\boldsymbol{g}_t)\right\|^2$$
$$= \mathbb{E}\left\|\boldsymbol{F}_{t-1} - (\boldsymbol{G}_{t-1})^\top \nabla f(\boldsymbol{g}_{t-1})\right\|^2$$
$$+ \mathbb{E}\left\| (\boldsymbol{G}_{t-1})^\top \nabla f(\boldsymbol{g}_{t-1}) - (\boldsymbol{G}_{t-1})^\top \nabla f_{i_t}(\boldsymbol{g}_{t-1}) + (\boldsymbol{G}_t)^\top \nabla f_{i_t}(\boldsymbol{g}_t) - (\boldsymbol{G}_t)^\top \nabla f(\boldsymbol{g}_t) \right\|^2$$
$$= \mathbb{E}\left\|\boldsymbol{F}_{t-1} - (\boldsymbol{G}_{t-1})^\top \nabla f(\boldsymbol{g}_{t-1})\right\|^2$$
$$+ \mathbb{E}\left( \left\| (\boldsymbol{G}_{t-1})^\top \nabla f(\boldsymbol{g}_{t-1}) - (\boldsymbol{G}_{t-1})^\top \nabla f_{i_t}(\boldsymbol{g}_{t-1}) + (\boldsymbol{G}_t)^\top \nabla f_{i_t}(\boldsymbol{g}_t) - (\boldsymbol{G}_t)^\top \nabla f(\boldsymbol{g}_t) \right\|^2\right)$$
$$\leq \mathbb{E}\left\|\boldsymbol{F}_{t-1} - (\boldsymbol{G}_{t-1})^\top \nabla f(\boldsymbol{g}_{t-1})\right\|^2 + \mathbb{E}\left\|(\boldsymbol{G}_t)^\top \nabla f_{i_t}(\boldsymbol{g}_t) - (\boldsymbol{G}_{t-1})^\top \nabla f_{i_t}(\boldsymbol{g}_{t-1})\right\|^2. \tag{41}$$

Applying (41) iteratively from $1$ to $t$ leads to (39).

(ii) We have

$$\mathbb{E}\left\|\boldsymbol{F}_t - (\boldsymbol{G}_t)^\top \nabla f(\boldsymbol{g}_t)\right\|^2$$
$$\overset{(a)}{\leq} \mathbb{E}\left\|\boldsymbol{F}_{\lfloor t/q\rfloor q} - (\boldsymbol{G}_{\lfloor t/q\rfloor q})^\top \nabla f(\boldsymbol{g}_{\lfloor t/q\rfloor q})\right\|^2 + \sum_{s=1}^{t} \mathbb{E}\left\|(\boldsymbol{G}_s)^\top \nabla f_{i_t}(\boldsymbol{g}_s) - (\boldsymbol{G}_{s-1})^\top \nabla f_{i_t}(\boldsymbol{g}_{s-1})\right\|^2$$
$$\overset{(b)}{\leq} \omega_3 + \sum_{s=1}^{t}\left(2M_f^2 \mathbb{E}\|\boldsymbol{G}_s - \boldsymbol{G}_{s-1}\|_F^2 + 2M_g^2 L_f^2 \mathbb{E}\|\boldsymbol{g}_s - \boldsymbol{g}_{s-1}\|^2\right)$$
$$\overset{(c)}{\leq} \omega_3 + 2\left(M_g^4 L_f^2 + M_f^2 L_g^2\right)\sum_{s=1}^{t}\mathbb{E}\|x_s - x_{s-1}\|^2$$
$$= \omega_3 + 2\left(M_g^4 L_f^2 + M_f^2 L_g^2\right)\cdot \eta^2 \sum_{s=1}^{t}\mathbb{E}\|\boldsymbol{F}_{s-1}\|^2, \tag{42}$$

where $(a)$ is due to (39), $(c)$ comes from $\boldsymbol{g}$-iteration and $\boldsymbol{G}$-iteration in Algorithm 1 as well as Assumptions 2 and 3 on smoothness and boundedness.

(iii) The only left is $(b)$, where we utilize (21) and note that for each $s = 1, \ldots, t$

$$\mathbb{E} \left\| (\boldsymbol{G}_s)^\top \nabla f_{i_t}(\boldsymbol{g}_s) - (\boldsymbol{G}_{s-1})^\top \nabla f_{i_t}(\boldsymbol{g}_{s-1}) \right\|^2$$

$$= \mathbb{E} \left\| (\boldsymbol{G}_s)^\top \nabla f_{i_t}(\boldsymbol{g}_s) - (\boldsymbol{G}_{s-1})^\top \nabla f_{i_t}(\boldsymbol{g}_s) + (\boldsymbol{G}_{s-1})^\top \nabla f_{i_t}(\boldsymbol{g}_s) - (\boldsymbol{G}_{s-1})^\top \nabla f_{i_t}(\boldsymbol{g}_{s-1}) \right\|^2$$

$$\leq 2\mathbb{E} \left\| (\boldsymbol{G}_{s-1})^\top \nabla f_{i_t}(\boldsymbol{g}_s) - (\boldsymbol{G}_{s-1})^\top \nabla f_{i_t}(\boldsymbol{g}_{s-1}) \right\|^2 + 2\mathbb{E} \left\| (\boldsymbol{G}_s)^\top \nabla f_{i_t}(\boldsymbol{g}_s) - (\boldsymbol{G}_{s-1})^\top \nabla f_{i_t}(\boldsymbol{g}_s) \right\|^2$$

$$\leq 2 \left( M_g^2 L_f^2 \mathbb{E} \left\| \boldsymbol{g}_s - \boldsymbol{g}_{s-1} \right\|^2 + M_f^2 \mathbb{E} \left\| \boldsymbol{G}_s - \boldsymbol{G}_{s-1} \right\|_F^2 \right),$$

$$\tag{43}$$

which result in $(b)$.

$$\square$$