[Reviews · NeurIPS 2019]

Reviewer 1



### I thank the authors for their feedback, in particular for clarification on notation and explaining that large minibatches should be estimated only every O(1/eps^2) steps, which is indeed rare. The other reviewers also convinced me that despite not having the right assumptions for the mention applications, the work might still be useful in other applications. I request the authors to remove the applications mentioned in the introduction or to explicitly write that their assumptions are not satisfied for them. Based on this points, I increase my score from 4 to 6. Let me also clarify on why I believe having the right assumption is important and what I dislike about the theory. SARAH is an interesting method as it does not require bounded gradients and, at the same time, there are settings where the its known complexity is better than that of SGD. However, here the requirements are quite restrictive and do not address applications mentioned in the introduction. The authors argue in the rebuttal that on the optimization trajectory the assumption should still hold, but I also have seen this argument for the method in [1], which was pointed out in [2] to provably diverge (with exponential rate!) on simple bilinear problems. Another metaphor is proximal gradient. Why do we need it if subgradient converges under bounded gradient assumption? It turns out that the assumptions actually matter and proximal gradient converges faster. So I believe that having reasonable assumption related to the problem motivation is important. [1] Solving variational inequalities with stochastic mirror-prox algorithm, Anatoli Juditsky, Arkadii S. Nemirovskii, Claire Tauvel [2] Reducing Noise in GAN Training with Variance Reduced Extragradient, Tatjana Chavdarova, Gauthier Gidel, Fran├žois Fleuret, Simon Lacoste-Julien ### 1. Table 1 is confusing because in Wang 2017a function g is allowed to be nonsmooth and in the current work it is assumed to be smooth (so the comparison is not fair). Would the authors agree to add columns to table 1 to show the difference in the assumptions? 2. It is very confusing that the authors write \partial g_j(x) instead of \nabla g_j(x) since they assume that g_j is smooth (equation 12). 3. Assumption 4 is not satisfied for function f in equation (4). For instance take w_1=w_2=...=w_N=0 and let w_{N+1}->+\infty. Then ||\nabla f||->+\infty. Similarly, it is not satisfied for f in line 335. Moreover, it seems that Assumption 9 is not satisfied for g_j in Risk management problem because g_j(x) - g(x) includes , which is not bounded and goes to infinity for x->\infty. 4. I like the guarantees for the finite sum case, but your minibatch size is 1/eps^2 in the online setting, which probably means that it's not going to be practical. 5. I see that you put a lot of efforts in deriving the guarantees, but it's unclear whether your work is going to help other researchers. Due to the high number of assumptions, it's more like a technical exercise of extending SARAH by assuming that every iteration is very close to a full gradient step. SARAH itself, in contrast, is very interesting because it works under very general assumptions. I didn't enjoy reading the proofs and feel that they give no new insight about optimization. Typos: line 26: it should be "at the same time" In equation (4) square is missing in the second term. line 77: "it adopt" should be "it adopts"

Reviewer 2



Update: I have read the authors response and I am satisfied the authors understand the issues I raised and will act to correct them. I would like two points for the authors to keep in mind when revising the paper: 1) Carefully explain your assumptions and the limitations they imply. In particular: - Reviewer 1 noted that some of the motivating applications in your paper don't satisfy the bounded gradient assumption. You should discuss this explicitly when presenting the application and/or the assumptions. - The nonstandard and misleading use of ||A|| to denote Frobenius norm (sometime without explicit definition!) appears pervasive in the whole line of work on compositional optimization. I urge you to stop this with your paper: use ||A||_F to denote Frobenius norm and explain clearly why you have to use this norm rather than operator norm, and how much this potentially harms your bounds. 2) Don't use the word "optimal" or "sharp" to describe your rates if you can't back it up. While your response acknowledges this error, it also repeats it the response to Reviewer 3 (line 46). As a rule of thumb, if you can't put a theorem or a citation next to "optimal" or "sharp", just don't write it. Claiming an improvement relative to previously known upper bounds is enough. =========================================================== This paper applies the stochastic recursive (SARAH) variance reduction technique to the nonconvex compositional optimization problem and improves the best known complexity bounds for it. The development consists of straightforward but clean application and analysis of SARAH for this problem. The paper also contains experiments but I did not review them (see second comment below). Overall, I believe this paper represent a respectable increment in the line of work on compositional optimization and is likely to interest a subset of the NeurIPS community. However, the paper suffers from a number of presentation issues which I detail below. 1) The use of Frobenius norm. Most works that use ||v|| to denote Euclidean norm of vector v use ||A|| to denote the Euclidean operator norm of matrix A. However, this paper apparently uses ||A|| to denote the Frobenius norm of the matrix; lines 88--89 state this vaguely, defining the notation ||A||_F and mysteriously confining the definition of Frobenius norm to square matrices. This is despite the fact that the operator is more natural for most of the analysis, and is obviously tighter. However, Lemma 7 crucially depends on using Frobenius rather the operator norm. The authors should clearly indicate their use of Frobenius norm using the notation ||A||_F. Moreover, the should explicitly discuss this caveat and its implications on complexity, and explain whether it also applies to the previous works. I would like to see this explantation in the rebuttal as well. 2) Experiments in Appendix. The authors include experiments with the proposed method in the appendix. However, the body of the paper refers to these experiments only in two sentences (the final sentence of the abstract and the final sentence of the introduction). I find this unacceptable: the body of the paper should fully describe its contribution. Moving contents completely to the appendix violates the author guidelines of some conferences similar to NeurIPS, e.g. COLT. Consequently, I decided not to review the experiments. I would suggest that the authors either remove the experiments from the paper completely, or include at least one description paragraph of the experiments and their findings in the body of the paper. 3) Inconsistent notation. The paper suffers from multiple instances of inconsistent and confusing notation. In particular: * F_{\xi} and f_{\xi} seem to mean the same thing, and the same goes for G_{\xi} and g_{\xi} (compare Eq. (7) to Algorithm 1 and Eq. (18) and (19)). This is particularly confusing because \boldsymbol{G}_t represents a sample of the Jacobian of g. * M_g, B_g and B_G all seem to mean the same thing. Ditto for M_f, B_f and B_F * The notation representing a random sample and random sample size is not clearly defined. 4) Vague optimality claims. The authors make a number of vague and unsubstantiated passive-voice claims about possible oracle optimality of the proposed method: in the abstract they call their complexity upper bound "sharp" without justification, and claim their result "is believed" to be optimal. In the introduction they claim that SARAH "is known" to be near optimal and "regarded as" cutting edge. None of these vague claims have room in a scientific paper. The authors should either change them to precise statement and back them up with precise reference or proof, or remove them altogether. 5) The paper contains a number of confusing mathematical typos. * In the (first) definition of \boldsymbol{F}_t in Algorithm 1, the multiplication by \boldsymbol{G}_t^T is missing * In Theorem 8 the second displayed equation has m instead of q. Moreover it should be q=O(m+n) rather than O(n). Also \mathcal{L}_1 should be \mathcal{C}_1. * In two places in Eq. (29) \xi_{t-1} should be \xi_t. This error repeats again and again in Appendix A. * Eq. (29) uses L_{\Phi} instead of L_f * In the proof of Theorem 10, I believe the Lipschitz constants in B_1 and C_1 may be wrong; please check them. A few additional comments: 6) It is not necessary to explicitly assume the Lipschitz constant L_\Phi, as it is implied by the rest of the Lipschitz assumption on g, f and their gradients. This also means that the constant C_* you define is always larger that L_\Phi. Consequently, the step size eta can be simplified by removing the maximum --- please do so. 7) The proof of theorem 10 does not rely on "standard concentration bounds;" it only relies on the formula for the variance of the average of iid vector random variable - please state this more precisely. (This part of the proof again relies on using Frobenius rather than operator matrix norms). 8) Typos. * Line 24: Risk -> The risk * line 56: rate with -> rate as * Line 75: a recently -> the recently * Line 111: remark -> remarking * Line 113: Assumptions -> assumptions * Line 124: integrated -> integrate * Line 148: put -> place * Line 151: of such -> of this case * Line 173: the only left -> the only transition left

Reviewer 3



=== after authors' rebuttal === Thanks for authors' response. After reading the rebuttal, I thank the authors answer my questions and would like to change my score from 4 to 6. === === === === === === === This study proposed a SARAH-type algorithm, SARAH-Compositional, for solving non-convex compositional optimization problems in both online and finite-sum settings. The proposed algorithms can enjoy improved convergence rates comparing with the-state-of-art results, and the main technique of analysis is borrowed from SARAH. This paper is easy to understand and well-organized. However, I have two major concerns. First, the novelty of this paper is limited since it only combines SARAH estimator with SGD. The analysis, in fact, is quite straightforward and trivial. Second, this paper didn't provide any numerical results, which are of great interest to many machine learning researchers.

[Author Response · NeurIPS 2019]

We thank all the reviewers for their valuable feedback. We appreciate the typos, minor errors, stylistic suggestions and unclear math steps pointed out by reviewers, and we will update our paper accordingly.We address the main comments from reviewers (abbreviated)

## Reviewer_1

**Q1: Unfair comparison with Wang 2017a because of smoothness assumption** A1: For earlier works (such as Wang 2017a) dealing with non-smooth cases, the algorithm using its techniques share the same rate with the smooth case itself. That being said, we will add columns to Table 1 to highlight the assumption difference.

**Q2: mixing $\partial g$ with $\nabla g$** A2: Thanks for pointing that out. We decide to keep this notation $\partial g$ for two reasons: (1) the notation has been adopted by a few earlier works (Lin et al 2018, Liu et al 2017) for smooth cases, where the "partial" notation is simply adopted to highlight that it is a (Jacobian) matrix; (2) our work is not addressing the non-smooth case, so there is no confusion with sub-gradient in the non-smooth case.

**Q3: Assumptions not satisfied for equation (4)** A3: First of all, (4) has a minor typo: the second sum should have its bracket squared. Although the derivative is not bounded, the differentiable functions $f$ and $g$ does admit bounded Lipschitz derivatives in the domain of optimization (a domain that contains the initializer, optimizer, and the entire path), and their compositional function also has Lipschitz derivatives. Our Assumptions 2, 3, 4 and 9 of Lipschitz class of functions are very similar to those in standard literature on compositional optimization (Lin et al 2018, Liu et al 2017).

**Q4: Mini-batch assumption of order $\mathcal{O}(1/(\epsilon^2))$ is impractical** A4: Different from $A_2 = B_2 = C_2 = 1$, the large-batch step $A_1, B_1, C_1$ occurs only once every $\epsilon^{-2}$ steps, so the IFO complexity remains unchanged. These large-batch steps helps reduce the variance of gradient estimation to $\mathcal{O}(\epsilon)$, and is very crucial for SARAH-type variance reduction of not accumulating noises.It is a selection which would allow the optimal theoretical guarantees (by central limit theorem, a $\mathcal{O}(1/(\epsilon^2))$ mini-batch every $q$ steps would give $\epsilon$-accurate estimator in the beginning).

**Q5: High number of assumptions and thus no new insight to help researchers** A5: The theoretical guarantees are valuable since they provide a state-of-the-art, and is believed to be optimal (although there is no lower bound result yet). As mentioned earlier, all assumptions are in standard literature and they see valuable applications in portfolio management, reinforcement learning, dimension reduction, etc.

## Reviewer_2

The reviewer is extremely careful in checking our proof. We appreciate that a lot. And we have fixed all the typos accordingly.

**Q1: Misleading use of the $\|\cdot\|$ as the Frobenius norm** A1: Thanks for your comment. Indeed, all norms involving the SARAH estimator for Jacobian matrices adopts a Frobenius norm. The complexity results still hold after a simple fix. Previous literatures also admit this "caveat" because there is potentially a gap between the Frobenius and (its lower-bounded) operator norms, this would not lead to any disadvantage of complexities.

**Q2: Experimental parts should be discribed in the body.** A2: We've done some experiments on this problem to validate our theory before the review period. We did more careful experimental setting and testing on the three applications mentioned: portfolio management, reinforcement learning, and t-SNE after then. We would position part of the experiments in the body part instead of the proof in our next submission.

**Q3: Inconsistent notation** A3: Thanks for checking in details. $F_\xi$ and $f_\xi$, $M_g$, $B_g$ and $B_G$ are indeed the same thing. We seperately used $f_i$ and $F_\xi$ before, which leads to duplicated constants. We already fixed them in our follow-up version.

**Q4: Optimality claims are vague** A4: We appreciate your preciseness. Our IFO complexity is state-of-the-art compared with previous literature. We will state our contribution in a more rigorous fashion.

**Q5: $L_\Phi$ is not necessary. Step size eta can be simplified** A5: We agree and already made this fix in our revision. Thank you.

**Q6: The proof of Thm 10 only relies on the formula for the variance of the average of iid vector random variable** A6: Yes, we corrected the statement of Theorem 10. Thanks for pointing that out

## Reviewer_3

**Q1: Novelty is limited** A1: We would like to argue our paper is not an incremental one. We believe that using variance reduced gradient methods (SVRG, SPIDER, among others) can be a potential alternative to existing stochastic compositional optimization methods, which includes and extends the current framework of SGD and SCGD. We aimed to provide an "optimal" nonconvex analysis and hence augment the current theoretical framework of this problem. We will try to polish more of this work in the next round of submission.

**Q2: Lacking numerical results** A2: Our work mainly focuses on proving that the convergence rates are sharper than all other existing convergence rates currently available for the compositional optimization problem. This demonstrates the power of using the SPIDER estimator to trace quantities needed. This work is mainly a theoretical extension along the directions pointed by the SPIDER paper. We did several experiments on three traditional applications in the field of compositional optimization problems. Our main body of the paper mainly focuses on proposing an optimal theoretical guarantee. See also A2 of **Reviewer_2**

[Meta-Review · NeurIPS 2019]

This paper has been deeply discussed between the reviewers and myself. After a lengthy discussion and thanks to the authors' rebuttal, the reviewers were convinced that the proposed algorithm and its analysis and novel, interesting, and worth to be published in NeurIPS. However, the reviewers also noted the mismatch between the motivating examples in the introduction and the assumptions in the analysis. This must be fixed. Note that it is not enough to state that the assumptions hold in the "domain of optimization" because there is no guarantee that such domain is bounded. So, please carefully take into account the reviewers' comments in preparing the camera-ready version.